# The Heat Transfer Problem in a Non-Convex Body—A New Procedure for Constructing Solutions

Rogério Martins Saldanha da Gama 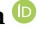

Mechanical Engineering Department, Universidade do Estado do Rio de Janeiro, São Fco Xavier Street 524, Rio de Janeiro 20550-013, Brazil; rsgama@terra.com.br

**Abstract:** The subject of this work is the coupled steady-state conduction-radiation-convection heat transfer phenomenon in a non-convex blackbody, which is represented by a second-order partial differential equation (representing the heat conduction inside the body) subjected to nonlinear (and non-local) boundary conditions (due to the thermal radiation heat transfer). Moreover, a non-convex body emits thermal radiant energy to itself, which must be taken into account in the boundary conditions when high temperatures are involved. The unknown is the absolute temperature. A procedure is proposed for constructing the solution to the problem by means of a sequence whose elements are obtained from linear problems, such as the classical ones involving linear Robin boundary conditions.

**Keywords:** non-convex blackbodies; thermal radiation; solution construction; sequence of linear problems

## 1. Introduction

Any real body at a temperature different from absolute zero emits thermal radiant energy, provided that it is surrounded by a non-opaque medium [1]. In general, at low temperatures, only the convection heat transfer between the body and the environment is taken into account, following the Newton law of cooling.

Nevertheless, when the body is surrounded by a rarefied atmosphere or when the temperature levels are high, the thermal radiation heat exchange must be taken into account, since it becomes a non-negligible mechanism for heat transfer between the body and its surroundings.

In addition, if this body is not convex, part of the thermal radiant energy emitted from its boundary reaches itself directly. When subsets of the body boundary are at high temperatures, the incident thermal radiant energy coming from the body boundary plays the role of a non-negligible external energy supply.

For instance, let us consider the part of a body represented in Figure 1. The points A and B can exchange, directly, thermal radiant energy.

In fact, any two points of body boundary, connectable by a straight line that does not intersect the body, exchange thermal radiant energy [1,2].

Since the Stefan–Boltzmann constant ($5.67 \times 10^{-8}$ watt per square meter per Kelvin to the fourth), the non-convexity effects may become negligible, especially for low temperatures and no rarefied atmospheres. Nevertheless, when temperatures greater than 300 Kelvin are involved and/or the vicinity is a rarefied atmosphere, the non-convexity effects may give rise to non-negligible contributions, since the emission from the body boundary to itself plays the role of a temperature dependent external source and the convective heat transfer becomes less effective.

Clearly, this is only an illustration of the non-convexity effect, but it serves to show the need of taking into account this effect.

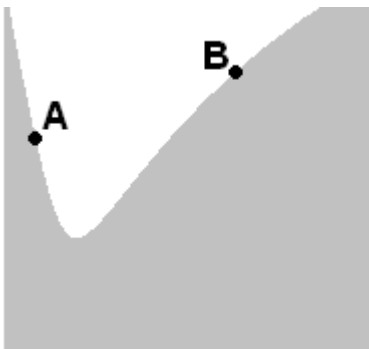

**Figure 1.** Example of a part of a non-convex body.

In order to take into account the thermal radiant energy exchange between points of the body boundary, the mathematical description becomes more complex. The boundary conditions will involve a relationship between the normal heat flux at each point on the boundary and the whole temperature distribution along this boundary.

The amount of thermal radiant energy exchanged between two points will depend on the local temperature as well as on the whole temperature distribution on some subsets of the body boundary.

The effect of the non-convexity on the heat transfer depends on the distance between the points on the boundary and on the angle between the normal vectors for each of the two points, provided a straight line can connect these points without passing inside the body (this line must be completely immersed in a non-opaque region). If the straight line connecting two points passes inside the body, then these points do not directly exchange thermal radiant energy. Convex bodies do not exhibitdirect thermal radiant energy interchange betweenpoints ontheir boundaries. Even so, the boundary conditions are non-linear due to the thermal radiation.

Assuming an opaque body at rest, a conduction heat transfer process takes place inside it, while a thermal radiation heat transfer and a convection heat transfer take place from/to the body boundary. It will be assumed the existence of known internal heat sources as well as known external thermal radiant sources [3].

In a non-convex body, part of the emitted thermal radiant energy reaches itself directly. In other words, a direct thermal radiant energy interchange takes place among points of the same body, even when these points are not neighboring [2,4].

The boundary condition for these problems arises naturally when the continuity of the normal heat flux on the boundary is assumed [5,6]. In other words, the normal conduction heat flux must be equal to the sum of both convection and thermal radiation heat flux at any point on the body boundary. Such a condition gives rise to a nonlinear boundary condition more complex than the usual ones employed in heat transfer problems [4].

Let us represent the body by the bounded open set $\Omega$. If $\Omega$ is not convex, then a part of the thermal radiant energy emitted from the body boundary will reach the body directly, playing the role of an external temperature dependent heat source. This effect takes into account the temperature distribution on a given subset of body boundary $\partial\Omega$.

Assuming the body to be rigid, opaque, and at rest, the energy transfer process inside $\Omega$ takes place by conduction heat transfer only. Hence, the energy transfer process considered here involves a coupling between a conduction heat transfer (inside $\Omega$), a thermal radiant heat transfer (from/to $\partial\Omega$), and a convection heat transfer (from/to $\partial\Omega$).

The main objective of this work is to construct the solution for the steady-state energy transfer process in a body (assumed black from a thermal radiant point of view) by means of a sequence whose elements are functions obtained from the solution of very simple and largely known linear heat transfer problems. Specifically, these well-known problems look like the classical conduction–convection heat transfer problems, in which the boundary

condition is a linear "Robin boundary condition," represented by the well-known Newton law of cooling, which can be found in most textbooks on heat transfer [7–10].

This type of problem is present in several situations in engineering, e.g., in the project of heating systems [11,12].

It is to be noticed that nonlinear heat transfer problems consist of an issue of permanent interest [13–17], in particular those involving nonlinear boundary conditions [18,19]. Nevertheless, the phenomenon considered in this work is little discussed in the current literature. In general, the boundary conditions employed here are poorly approximated by most authors (in order to avoid complex calculations).

## 2. The Mathematical Modeling

The steady-state conduction heat transfer process inside a rigid and opaque body at rest, represented by the set $\Omega$, is mathematically described as [6,9]

$$div\ (k\ \nabla T) + \dot{q} = 0\ in\ \Omega \tag{1}$$

where $T$ represents the temperature, $\dot{q}$ is a non-negative field (an internal heat source), and $k$ is the thermal conductivity (always positive valued). In this work, $\dot{q}$ and $k$ are assumed to be known, piecewise continuous, and bounded. In addition, $\Omega$ is assumed to possess the cone property, while $\partial\Omega$ is piecewise smooth.

Assuming a blackbody behavior, the thermal radiant energy (per unit time and per unit area) emitted from a point on the boundary $\partial\Omega$ is given by [20]

$$e_{EMIT} = \sigma|T|^3 T\ on\ \partial\Omega \tag{2}$$

where $\sigma$ is the Stefan–Boltzmann constant. The use of $|T|^3 T$ instead of $T^4$ is mathematically convenient and physically equivalent [20]. From a physical point of view, $T$ does not make sense if negative valued.

The incident thermal radiant energy (per unit time and per unit area) at a given point $\mathbf{x} \in \partial\Omega$ is given by [1,10]

$$e_{INC} = \int_{\mathbf{y}\in\partial\Omega} \sigma\left|\hat{T}(\mathbf{y})\right|^3 \hat{T}(\mathbf{y})K(\mathbf{x},\mathbf{y})dS + \hat{s}(\mathbf{x}),\ for\ all\ \mathbf{x} \in \partial\Omega \tag{3}$$

Equation (3) takes into account an external thermal radiant source, represented by $s = \hat{s}(\mathbf{x})$ (a known non-negative valued bounded function)as well as the effect of the thermal radiation that, emerging from points on $\partial\Omega$, reaches the point $\mathbf{x} \in \partial\Omega$. In (2) and (3), $T = \hat{T}(\mathbf{x})$ represents the absolute temperature at the point $\mathbf{x} \in \partial\Omega$.

Since the radiation emitted from a blackbody is diffusely distributed, the kernel $K(\mathbf{x},\mathbf{y})$ ($K(\mathbf{x},\mathbf{y}) \equiv K(\mathbf{y},\mathbf{x})$) depends only on the geometry of $\Omega$ [1] and is such that

$$0 \leq \int_{\mathbf{y}\in\partial\Omega} K(\mathbf{x},\mathbf{y})dS = \hat{\eta}(\mathbf{x}) < \mu \leq 1,\ for\ all\ \mathbf{x} \in \partial\Omega \tag{4}$$

Here, we admit that any point on the body boundary can emit thermal radiant energy directly to the environment in such a way that the non-negative constant $\mu$ is less than one. In fact, this is a sufficient condition for the protocol to be proposed here, not a necessary one.

Combining Equations (2) and (3), we have the thermal radiant heat flux on $\partial\Omega$ given by [1]

$$q_{RADIATION} = e_{EMIT} - e_{INC}\ on\ \partial\Omega \tag{5}$$

The convection heat transfer from/to body boundary is given by the Newton law of cooling [4,5] as

$$q_{CONVECTION} = h(T - T_\infty)\ on\ \partial\Omega \tag{6}$$

where $h$ and $T_\infty$ are known positive-valued functions (bounded and, in general, assumed constants).

The conduction heat flux on the boundary is given by (Fourier law) [6,9]

$$q_{CONDUCTION} = -k\nabla T \cdot \mathbf{n} \ on \ \partial\Omega \tag{7}$$

In order to ensure that there is no jump in the normal heat flux across the boundary, we must equal the normal conduction heat flux and the thermal radiant heat flux on $\partial\Omega$. With this aim in mind, we have

$$q_{CONDUCTION} = q_{RADIATION} + q_{CONVECTION} \ on \ \partial\Omega \ \Rightarrow$$
$$\Rightarrow \ -k\nabla T \cdot \mathbf{n} = e_{EMIT} - e_{INC} + h(T - T_\infty) \ on \ \partial\Omega \tag{8}$$

Combining (1) and (9), the resulting mathematical description for the steady-state heat transfer process yields

$$div(k\nabla T) + \dot{q} = 0 \ in \ \Omega$$
$$-k\nabla T \cdot \mathbf{n} = \sigma|T|^3 T - \Theta\left[\sigma|T|^3 T\right] - s + h(T - T_\infty) \ on \ \partial\Omega \tag{9}$$

where the unknown is the absolute temperature field $T$. The linear operator $\Theta$ is defined as

$$\Theta[\phi] = \int_{\mathbf{y}\in\partial\Omega} \hat{\phi}(\mathbf{y})K(\mathbf{x},\mathbf{y})dS, \ \phi = \hat{\phi}(\mathbf{x}), \ for \ all \ \mathbf{x} \in \partial\Omega \tag{10}$$

It is worth noting that $T$ is continuous in $\Omega$ and piecewise continuous on $\partial\Omega$.

## 3. Constructing the Sequence

Let us consider now the sequence $\left[\psi^1, \psi^2, \psi^3, \ldots\right]$ whose elements are obtained from the solution of the following problem:

$$div\left(k\nabla \psi^{i+1}\right) + \dot{q} = 0 \ in \ \Omega$$
$$-k\nabla \psi^{i+1} \cdot \mathbf{n} = \alpha\psi^{i+1} - \beta^i \ on \ \partial\Omega \tag{11}$$
$$with \ \beta^i = \alpha\psi^i - \sigma|\psi^i|^3\psi^i + \Theta\left[\sigma|\psi^i|^3\psi^i\right] + s - h\left(\psi^i - T_\infty\right) \ on \ \partial\Omega$$

where $\alpha$ is a sufficiently large positive constant and $\beta^i$ is (for each $i$) a known function. As already pointed out, the quantities $\dot{q}$ and $s$ are known non-negative valued functions. The thermal conductivity $k$ is always positive-valued. It is obvious that it is possible to define a new function as the sum $s + hT_\infty$, but this is not convenient from a physical point of view, since the natures of the terms arequite different.

The sequence $\left[\psi^1, \psi^2, \psi^3, \ldots\right]$ is obtained assuming that $\psi^0 \equiv 0$. The solution of problem (9) is given by

$$T \equiv \psi^\infty \equiv \lim_{i\to\infty}\psi^i \ in \ \Omega \tag{12}$$

as it will be shown later (see Equation (55)).

Taking into account that $\psi^0 \equiv 0$, the element $\psi^1$ is the solution of

$$div\left(k\nabla \psi^1\right) + \dot{q} = 0 \ in \ \Omega$$
$$-k\nabla \psi^1 \cdot \mathbf{n} = \alpha\psi^1 - s - hT_\infty \ on \ \partial\Omega \tag{13}$$

and, therefore, since $\dot{q} \geq 0$, $s + hT_\infty \geq 0$, and $\alpha > 0$, $\psi^1$ is non-negative valued everywhere [21,22].

## 4. On the Behavior of the Sequence for Sufficiently Large $\alpha$

The first step for proving that (12) holds is to show that the sequence $\left[\psi^1, \psi^2, \psi^3, \ldots\right]$ is nondecreasing. In order to show this, let us consider (11) for two consecutive elements of

the sequence. Taking into account that $h$, $T_\infty$, $k$, $\dot{q}$, and $s$ do not depend on the unknowns, we have

$$
\begin{aligned}
&div\big(k\nabla\big(\psi^{i+1} - \psi^i\big)\big) = 0 \ in \ \Omega \\
&-k\nabla\big(\psi^{i+1} - \psi^i\big)\cdot\mathbf{n} = \alpha\big(\psi^{i+1} - \psi^i\big) - \big(\beta^i - \beta^{i-1}\big) \ on \ \partial\Omega \\
&\beta^i - \beta^{i-1} = \\
&= (\alpha - h)\big(\psi^i - \psi^{i-1}\big) - \sigma\Big(\big|\psi^i\big|^3\psi^i - \big|\psi^{i-1}\big|^3\psi^{i-1}\Big) + \sigma\,\Theta\Big[\big|\psi^i\big|^3\psi^i - \big|\psi^{i-1}\big|^3\psi^{i-1}\Big] \ on \ \partial\Omega
\end{aligned}
\tag{14}
$$

The existence of a positive constant $\delta$ such that $k \geq \delta$ everywhere [4,7,8] enables us to conclude that

$$
\sup_{\partial\Omega}\big(\psi^{i+1} - \psi^i\big) = \sup_\Omega\big(\psi^{i+1} - \psi^i\big) \ and \ \inf_{\partial\Omega}\big(\psi^{i+1} - \psi^i\big) = \inf_\Omega\big(\psi^{i+1} - \psi^i\big) \tag{15}
$$

So, there exists a nonempty subset $\partial\Omega_{i+1}^-$, defined as follows [19,23],

$$
\partial\Omega_{i+1}^- \equiv \Big\{\mathbf{x} \in \partial\Omega, \ such \ that \ k\nabla\big(\psi^{i+1} - \psi^i\big)\cdot\mathbf{n} \leq 0\Big\}
$$

such that

$$
\inf_{\partial\Omega_{i+1}^-}\big(\psi^{i+1} - \psi^i\big) = \inf_{\partial\Omega}\big(\psi^{i+1} - \psi^i\big) = \inf_\Omega\big(\psi^{i+1} - \psi^i\big) \tag{16}
$$

From the boundary conditions on $\partial\Omega_{i+1}^-$ we have [24]

$$
\begin{aligned}
&\alpha\big(\psi^{i+1} - \psi^i\big) \geq \big(\beta^i - \beta^{i-1}\big) \ on \ \partial\Omega_{i+1}^- \ \Rightarrow \ \alpha\big(\psi^{i+1} - \psi^i\big) \geq \\
&\geq (\alpha - h)\big(\psi^i - \psi^{i-1}\big) - \sigma\Big(\big|\psi^i\big|^3\psi^i - \big|\psi^{i-1}\big|^3\psi^{i-1}\Big) + \sigma\,\Theta\Big[\big|\psi^i\big|^3\psi^i - \big|\psi^{i-1}\big|^3\psi^{i-1}\Big] \ on \ \partial\Omega_{i+1}^-
\end{aligned}
\tag{17}
$$

Taking into account that $\psi^1 \geq 0$ everywhere and that $\psi^0 \equiv 0$, we can write

$$
\begin{aligned}
&\alpha\big(\psi^2 - \psi^1\big) \geq \\
&\geq (\alpha - h)\big(\psi^1 - \psi^0\big) - \sigma\Big(\big|\psi^1\big|^3\psi^1 - \big|\psi^0\big|^3\psi^0\Big) + \sigma\,\Theta\Big[\big|\psi^1\big|^3\psi^1 - \big|\psi^0\big|^3\psi^0\Big] \ on \ \partial\Omega_2^-
\end{aligned}
\tag{18}
$$

Therefore, for a sufficiently large constant $\alpha$, the right-hand side is nonnegative on $\partial\Omega_2$, and we ensure, from (18), that $\psi^2 \geq \psi^1$ everywhere.

The above procedure may be repeated, giving rise to the following inequality

$$
\ldots \geq \psi^{i+1} \geq \psi^i \geq \ldots \geq \psi^2 \geq \psi^1 \geq \psi^0 \equiv 0 \ in \ \Omega \tag{19}
$$

provided, for any $i > 0$,

$$
(\alpha - h)\big(\psi^i - \psi^{i-1}\big) \geq \sigma\Big(\big|\psi^i\big|^3\psi^i - \big|\psi^{i-1}\big|^3\psi^{i-1}\Big) - \sigma\,\Theta\Big[\big|\psi^i\big|^3\psi^i - \big|\psi^{i-1}\big|^3\psi^{i-1}\Big] \ on \ \partial\Omega_{i+1}^- \tag{20}
$$

Any positive constant $\alpha$ such that

$$
\alpha\big(\psi^i - \psi^{i-1}\big) \geq \sigma\Big(\big|\psi^i\big|^3\psi^i - \big|\psi^{i-1}\big|^3\psi^{i-1}\Big) + h\big(\psi^i - \psi^{i-1}\big) \ on \ \partial\Omega, \ i = 1, 2, 3, \ldots \tag{21}
$$

ensures (20).

In order to establish a sufficiently large value for $\alpha$, we must obtain an upper bound for the sequence $\big[\psi^0, \psi^1, \psi^2, \psi^3, \ldots\big]$. As it will be shown later, the temperature $T$, solution of (9), is an upper bound for this sequence. So, any $\alpha$ such that

$$
\alpha \geq 4\sigma \sup_{\partial\Omega}\big\{|T|^3\big\} + \sup_{\partial\Omega}\{h\} \tag{22}
$$

satisfies (21) and ensures convergence.

Inequality (22) is a sufficient condition, not a necessary condition. So, a "trial and error" procedure may be used for choosing a convenient value for $\alpha$.

## 5. On the Convergence of the Sequence $[\psi^0, \psi^1, \psi^2, \psi^3, \ldots]$

From problem (14), we may write

$$\int_{\partial\Omega} \left\{ \alpha\left(\psi^{i+1} - \psi^i\right) - \left(\beta^i - \beta^{i-1}\right) \right\} dA = 0 \tag{23}$$

whichleads to

$$
\begin{aligned}
\int_{\partial\Omega} \alpha\left(\psi^{i+1} - \psi^i\right) dA = \\
= \int_{\partial\Omega} \left\{ (\alpha - h)(\psi^i - \psi^{i-1}) - \sigma\left(\left|\psi^i\right|^3 \psi^i - \left|\psi^{i-1}\right|^3 \psi^{i-1}\right) + \sigma\Theta\left[\left|\psi^i\right|^3 \psi^i - \left|\psi^{i-1}\right|^3 \psi^{i-1}\right] \right\} dA
\end{aligned}
\tag{24}
$$

Since

$$
\begin{aligned}
\int_{\partial\Omega} \sigma\,\Theta\left[\left|\psi^i\right|^3 \psi^i - \left|\psi^{i-1}\right|^3 \psi^{i-1}\right] dA = \\
= \int_{\mathbf{x}\in\partial\Omega} \left\{ \int_{\mathbf{y}\in\partial\Omega} \sigma\left\{ \left|\hat{\psi}^i(\mathbf{y})\right|^3 \hat{\psi}^i(\mathbf{y}) - \left|\hat{\psi}^{i-1}(\mathbf{y})\right|^3 \hat{\psi}^{i-1}(\mathbf{y}) \right\} K(\mathbf{x}, \mathbf{y}) dA \right\} = \\
= \int_{\mathbf{y}\in\partial\Omega} \sigma\left\{ \left|\hat{\psi}^i(\mathbf{y})\right|^3 \hat{\psi}^i(\mathbf{y}) - \left|\hat{\psi}^{i-1}(\mathbf{y})\right|^3 \hat{\psi}^{i-1}(\mathbf{y}) \right\} \left\{ \int_{\mathbf{x}\in\partial\Omega} K(\mathbf{x}, \mathbf{y}) dA \right\} dA \leq \\
\leq \mu \int_{\mathbf{y}\in\partial\Omega} \sigma\left\{ \left|\hat{\psi}^i(\mathbf{y})\right|^3 \hat{\psi}^i(\mathbf{y}) - \left|\hat{\psi}^{i-1}(\mathbf{y})\right|^3 \hat{\psi}^{i-1}(\mathbf{y}) \right\} dA
\end{aligned}
\tag{25}
$$

we ensure that

$$
\begin{aligned}
\int_{\partial\Omega} \left\{ -\sigma\left(\left|\psi^i\right|^3 \psi^i - \left|\psi^{i-1}\right|^3 \psi^{i-1}\right) + \sigma\Theta\left[\left|\psi^i\right|^3 \psi^i - \left|\psi^{i-1}\right|^3 \psi^{i-1}\right] \right\} dA \leq \\
\leq -(1 - \mu)\int_{\partial\Omega} \sigma\left(\left|\psi^i\right|^3 \psi^i - \left|\psi^{i-1}\right|^3 \psi^{i-1}\right) dA \leq 0
\end{aligned}
\tag{26}
$$

Therefore,

$$\int_{\partial\Omega} \alpha\left(\psi^{i+1} - \psi^i\right) dA \leq \int_{\partial\Omega} (\alpha - h)\left(\psi^i - \psi^{i-1}\right) dA \tag{27}$$

Defining the norm $\|\bullet\|$ as follows (the $L^1(\partial\Omega)$ norm) [25]

$$\|\bullet\| = \int_{\partial\Omega} |\bullet| dA \tag{28}$$

and taking into account inequality (21), we have

$$\left\|\psi^{i+1} - \psi^i\right\| \leq \left(1 - \frac{\overline{h}}{\alpha}\right)\left\|\psi^i - \psi^{i-1}\right\|, \quad \overline{h} = \min_{\partial\Omega} h > 0 \tag{29}$$

Clearly, when $h$ is assumed constant (most usual), $\overline{h} = h$.

Inequality (31) characterizes a contraction and ensures the convergence in the norm ($L^1(\partial\Omega)$) defined by (28). In addition, since $\psi^0 \equiv 0$, we may write,

$$\left\|\psi^{i+1} - \psi^i\right\| \leq \left(1 - \frac{\overline{h}}{\alpha}\right)^i \left\|\psi^1\right\| \tag{30}$$

In this way, we can define the limit of the sequence, denoted here as $\psi^\infty$, as the solution of the problem below

$$
\begin{aligned}
&\nabla \cdot (k\nabla \psi^\infty) + \dot{q} = 0 \; in \; \Omega \\
&-k\nabla \psi^\infty \cdot \mathbf{n} = \alpha \psi^\infty - \beta^\infty \; on \; \partial\Omega \\
&with \; \beta^\infty = \lim_{i\to\infty} \left\{ \alpha \psi^i - \sigma |\psi^i|^3 \psi^i + \Theta \left[ \sigma |\psi^i|^3 \psi^i \right] + s - h(\psi^i - T_\infty) \right\} \; on \; \partial\Omega
\end{aligned}
\tag{31}
$$

Hence, the regularity of $\psi^\infty$ in $\Omega$ is the same as $T$ (the solution of (9)), lending support to (12). In other words, it is proven that the limit of the sequence is the solution of the problem.

## 6. An Error Estimate

Combining (9) and (11), we have

$$
\begin{aligned}
div\left(k\nabla\left(T - \psi^{i+1}\right)\right) &= 0 \; in \; \Omega \\
-k\nabla\left(T - \psi^{i+1}\right)\cdot\mathbf{n} &= \\
= \sigma\left(|T|^3 T - |\psi^i|^3\psi^i\right) - \sigma\,\Theta\left[|T|^3 T - |\psi^i|^3\psi^i\right] &- \alpha\left(\psi^{i+1} - \psi^i\right) + h\left(T - \psi^i\right) \; on \; \partial\Omega
\end{aligned}
\tag{32}
$$

Defining the nonempty subset $\partial\Omega_{i+1}^{--}$ as follows

$$
\partial\Omega_{i+1}^{--} \equiv \left\{ \mathbf{x} \in \partial\Omega, \; such \; that \; k\nabla\left(T - \psi^{i+1}\right)\cdot\mathbf{n} \le 0 \right\}
$$

we can write

$$
\inf_{\partial\Omega_{i+1}^{--}} \left(T - \psi^{i+1}\right) = \inf_{\partial\Omega}\left(T - \psi^{i+1}\right) = \inf_{\Omega}\left(T - \psi^{i+1}\right)
\tag{33}
$$

On the subset $\partial\Omega_{i+1}^{--}$ we have

$$
\sigma\left(|T|^3 T - |\psi^i|^3\psi^i\right) - \sigma\,\Theta\left[|T|^3 T - |\psi^i|^3\psi^i\right] - \alpha\left(\psi^{i+1} - \psi^i\right) + h\left(T - \psi^i\right) \ge 0
\tag{34}
$$

The above inequality may be rewritten as

$$
\begin{aligned}
\sigma\left(|T|^3 T - \sigma|\psi^{i+1}|^3\psi^{i+1}\right) - \sigma\,\Theta\left[|T|^3 T - |\psi^i|^3\psi^i\right] + h\left(T - \psi^{i+1}\right) &\ge \\
\ge (\alpha - h)\left(\psi^{i+1} - \psi^i\right) - \sigma\left(|\psi^{i+1}|^3\psi^{i+1} - |\psi^i|^3\psi^i\right) \; on \; \partial\Omega_{i+1}^{--}
\end{aligned}
\tag{35}
$$

Since, from (21),

$$
(\alpha - h)\left(\psi^{i+1} - \psi^i\right) - \sigma\left(|\psi^{i+1}|^3\psi^{i+1} - |\psi^i|^3\psi^i\right) \ge 0 \; on \; \partial\Omega, \; i = 1, 2, 3, \ldots
\tag{36}
$$

inequality (37) gives rise to

$$
\sigma\left(|T|^3 T - \sigma|\psi^{i+1}|^3\psi^{i+1}\right) - \sigma\,\Theta\left[|T|^3 T - |\psi^i|^3\psi^i\right] + h\left(T - \psi^{i+1}\right) \ge 0 \; on \; \partial\Omega_{i+1}^{--}
\tag{37}
$$

This inequality ensures that, if $T \ge \psi^i$ on $\partial\Omega$, then $T \ge \psi^{i+1}$ on $\partial\Omega_{i+1}^{--}$. Therefore, since $\psi^1 \ge \psi^0 \equiv 0$, we are able to conclude that

$$
\inf_{\partial\Omega_{i+1}^{--}} \left(T - \psi^{i+1}\right) = \inf_{\partial\Omega}\left(T - \psi^{i+1}\right) = \inf_{\Omega}\left(T - \psi^{i+1}\right) \ge 0 \; i = 1, 2, 3, \ldots
\tag{38}
$$

In other words, $T \ge \psi^i \; in \; \Omega$ ($T$ is an upper bound for the sequence $\left[\psi^0, \psi^1, \psi^2, \psi^3, \ldots\right]$). It is possible to establish an error estimate for each element of the sequence with respect to the exact solution $T$. In order to do this, let us integrate the boundary condition of problem (34), yielding

$$\int_{\partial\Omega} \left\{ \sigma\left( |T|^3 T - \left|\psi^i\right|^3 \psi^i \right) - \sigma\,\Theta\left[ |T|^3 T - \left|\psi^i\right|^3 \psi^i \right] - \alpha\left( \psi^{i+1} - \psi^i \right) + h\left( T - \psi^i \right) \right\} dA = 0 \tag{39}$$

Taking into account that $T \geq \psi^i$ and considering (4), we have

$$\int_{\partial\Omega} \left\{ \sigma(1 - \mu)\left( |T|^3 T - \left|\psi^i\right|^3 \psi^i \right) + h\left( T - \psi^i \right) \right\} dA \leq \int_{\partial\Omega} \alpha\left( \psi^{i+1} - \psi^i \right) dA \tag{40}$$

In addition, since

$$|T|^3 T - \left|\psi^i\right|^3 \psi^i \geq \left( T - \psi^i \right)^4 \tag{41}$$

inequality (41) yields (taking into account Schwarz inequality [23])

$$\frac{\sigma(1 - \mu)}{A^3} \left( \int_{\partial\Omega} \left( T - \psi^i \right) dA \right)^4 + h \int_{\partial\Omega} \left( T - \psi^i \right) dA \leq \int_{\partial\Omega} \alpha\left( \psi^{i+1} - \psi^i \right) dA \tag{42}$$

in which $A$ is the area of $\partial\Omega$.

Inequality (42) gives rise to the following error estimate

$$\left\| T - \psi^i \right\| \leq MIN\left( \frac{\alpha A^3}{\sigma(1 - \mu)} \left\| \psi^{i+1} - \psi^i \right\|^{1/4}, \frac{\alpha}{\underline{h}} \left\| \psi^{i+1} - \psi^i \right\| \right), \quad \|\bullet\| = \int_{\partial\Omega} |\bullet| dA \tag{43}$$

It is remarkable that the supremum of $T - \psi^i$ on $\partial\Omega$ coincides with the supremum of $T - \psi^i$ in $\Omega$. The same holds for the difference $\psi^{i+1} - \psi^i$.

## 7. An a Priori Upper Bound for *T*

The knowledge of an upper bound estimate is always a useful bit of information [26] and, in this work, may be used for estimating a sufficient value for the constant $\alpha$. Consider again problem (5) and the following inequality (there are infinitely many choices for $v$)

$$div(k\nabla v) + \dot{q} \leq 0 \; in \; \Omega \tag{44}$$

So,

$$div(k\nabla(T - v)) \geq 0 \; in \; \Omega$$
$$-k\nabla(T - v)\cdot\mathbf{n} = \sigma|T|^3 T - \Theta\left[ \sigma|T|^3 T \right] + h(T - T_\infty) - s + k\nabla v\cdot\mathbf{n} \; on \; \partial\Omega \tag{45}$$

At this point, we introduce the nonempty subset $\partial\Omega^{\#}$, defined as follows

$$\partial\Omega^{\#} \equiv \{\mathbf{x} \in \partial\Omega, \; such \; that \; k\nabla(T - v)\cdot\mathbf{n} \geq 0\} \tag{46}$$

Therefore,

$$\sup_{\partial\Omega^{\#}}(T - v) = \sup_{\partial\Omega}(T - v) = \sup_{\Omega}(T - v) \tag{47}$$

Hence, we have

$$\sigma|T|^3 T - \Theta\left[ \sigma|T|^3 T \right] + h(T - T_\infty) - s + k\nabla v\cdot\mathbf{n} \leq 0 \; on \; \partial\Omega^{\#} \tag{48}$$

This inequality yields

$$\sigma|T|^3 T + hT \leq \Theta\left[ \sigma|T|^3 T \right] + hT_\infty + s + \sup_{\partial\Omega}\|k\nabla v\| \; on \; \partial\Omega^{\#} \tag{49}$$

and gives rise to

$$
\begin{aligned}
\sigma(1-\mu)\sup_{\partial\Omega^{\#}}\left(|T|^{3}T\right) &\leq \sup_{\partial\Omega^{\#}}(hT_{\infty}+s)+\sup_{\partial\Omega}\|k\nabla v\| \ and\\
\overline{h}\sup_{\partial\Omega^{\#}}(T) &\leq \sup_{\partial\Omega^{\#}}(hT_{\infty}+s)+\sup_{\partial\Omega}\|k\nabla v\|
\end{aligned}
\tag{50}
$$

Taking into account that $\partial\Omega^{\#}\subset\partial\Omega$, we have

$$
\sup_{\partial\Omega^{\#}}(T-v)\leq \sup_{\partial\Omega^{\#}}(T)-\inf_{\partial\Omega}(v)
\tag{51}
$$

Since

$$
\sup_{\Omega}(T-v)\leq \sup_{\partial\Omega^{\#}}(T)-\inf_{\partial\Omega}(v) \ \Rightarrow \ \sup_{\Omega}(T)\leq \sup_{\partial\Omega^{\#}}(T)-\inf_{\partial\Omega}(v)+\sup_{\Omega}(v)
\tag{52}
$$

Therefore, we are able to establish an a priori upper bound for $T$, with the aid of (51). With this upper bound estimate, we may estimate $\alpha$ from (22). The following choices are always valid, and we can choose the least value between the ones below

$$
\sup_{\partial\Omega^{\#}}(T)\leq \left\{ \frac{\sup_{\partial\Omega^{\#}}(hT_{\infty}+s)+\sup_{\partial\Omega}\|k\nabla v\|}{\sigma(1-\mu)} \right\}^{1/4} \ or \ \sup_{\partial\Omega^{\#}}(T)\leq \left\{ \frac{\sup_{\partial\Omega^{\#}}(hT_{\infty}+s)+\sup_{\partial\Omega}\|k\nabla v\|}{\overline{h}} \right\}
\tag{53}
$$

Nevertheless, this calculation serves, basically, to show that there exists a value for $\alpha$ that ensures the sequence isnon-decreasing and convergent. In fact, the upper bound estimate yields very large values of $\alpha$, giving rise to low convergence speeds.

It is recommended that the "trial and error" procedure be used. If the choice of $\alpha$ does not provide a non-decreasing sequence, then we increase $\alpha$.

Since the elements of the sequence as well as $T$ are bounded and continuous in $\Omega$, the limit defined in (12) can be regarded in a Sobolev sense ($H^{1}(\Omega)$). In other words,

$$
\lim_{i\to\infty}\left\|T-\psi^{i}\right\|_{H^{1}(\Omega)}=0
\tag{54}
$$

## 8. A One Dimensional Example—Spherical Shell

In order to illustrate the proposed procedure, let us consider a very simple heat transfer problem, involving a spherical shell (as suggested in Figure 2) in which the temperature depends only on the radial variable, $s\equiv 0$, $k$ = constant, $h$ = constant, $T_{\infty}$ = constant, and $\dot{q}$ = constant.

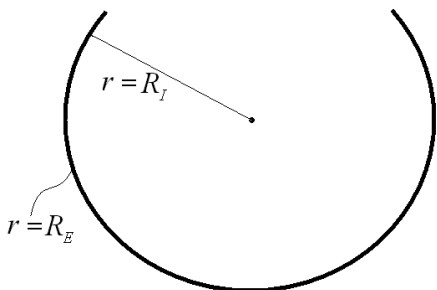

**Figure 2.** The studied problem. Spherical shell with internal radius $R_{I}$ and external radius $R_{E}$.

In this case, the steady-state heat transfer problem has the mathematical description given by

$$k\, div\nabla T + \dot{q} = 0\ in\ \Omega$$
$$-k\nabla T\cdot\mathbf{n} = \sigma|T|^3 T - \Theta\Big[\sigma|T|^3 T\Big] + h(T - T_\infty)\ on\ \partial\Omega \tag{55}$$

where the set $\Omega$ is defined in terms of the radial variable as $R_I < r < R_E$, while the boundary consists of the points $r = R_I$ and $r < R_E$.

Here, the kernel is given by [1]

$$K(\mathbf{x}, \mathbf{y}) = \begin{cases} \frac{1}{4\pi R_I^2} = constant\ , & r = R_I \\ 0\ , & r = R_E \end{cases} \tag{56}$$

Problem (56) may be rewritten as

$$\frac{1}{r^2}\frac{d}{dr}\left(r^2\frac{dT}{dr}\right) + \frac{\dot{q}}{k} = 0,\ R_I < r < R_E$$
$$k\frac{dT}{dr} = \sigma|T|^3 T - \left(\frac{A}{4\pi R_I^2}\right)\sigma|T|^3 T + h(T - T_\infty),\ r = R_I,\ 0 < \frac{A}{4\pi R_I^2} < 1 \tag{57}$$
$$-k\frac{dT}{dr} = \sigma|T|^3 T + h(T - T_\infty),\ r = R_E$$

where $A$ is the area of the internal spherical surface $r = R_I$. The solution has the form

$$T = -\frac{\dot{q}r^2}{6k} + \frac{C_1}{r} + C_2,\ R_I \le r \le R_E \tag{58}$$

Defining the quantities

$$\gamma = \frac{A}{4\pi R_I^2},\ a = \frac{\sigma R_I^7 \dot{q}^3}{k^4},\ b = \frac{hR_I}{k},\ \theta_\infty = \frac{kT_\infty}{\dot{q}R_I^2},\ \delta = \frac{R_E}{R_I} - 1,\ \theta = \frac{kT}{\dot{q}R_I^2}\ and\ \xi = \frac{r}{R_I} \tag{59}$$

we obtain a dimensionless form of the problem

$$\frac{1}{\xi^2}\frac{d}{d\xi}\left(\xi^2\frac{d\theta}{d\xi}\right) + 1 = 0,\ 1 < \xi < 1 + \delta$$
$$\frac{d\theta}{d\xi} = a|\theta|^3\theta - \gamma a|\theta|^3\theta + b(\theta - \theta_\infty),\ \xi = 1 \tag{60}$$
$$-\frac{d\theta}{d\xi} = a|\theta|^3\theta + b(\theta - \theta_\infty),\ \xi = 1 + \delta$$

with the following solution

$$\theta = -\frac{\xi^2}{6} + \frac{C_1}{\xi} + C_2,\ 1 \le \xi \le 1 + \delta \tag{61}$$

that can be written as

$$\theta = -\frac{\xi^2}{6} + \frac{1}{6}\left\{-\frac{6\theta_E(1+\delta) + (1+\delta)^3}{\delta}\left(\frac{1}{\xi} - 1\right) + \frac{6\theta_I + 1}{\delta}\left(\frac{1+\delta}{\xi} - 1\right)\right\},\ 1 \le \xi \le 1 + \delta \tag{62}$$

where $\theta_I$ represents the dimensionless temperature $\theta$ at the position $\xi = 1$, while $\theta_E$ represents the dimensionless temperature $\theta$ at the position $\xi = 1 + \delta$.

Now, let us consider the sequence associated with the above (dimensionless) problem. Its elements are a solution of

$$
\begin{aligned}
&\frac{1}{\xi^2}\frac{d}{d\xi}\left(\xi^2\frac{d\psi^{i+1}}{d\xi}\right)+1=0,\ 1<\xi<1+\delta\\
&\frac{d\psi^{i+1}}{d\xi}=\alpha\psi^{i+1}-\beta_I^i,\ \xi=1\\
&-\frac{d\psi^{i+1}}{d\xi}=\alpha\psi^{i+1}-\beta_E^i,\ \xi=1+\delta\\
&\beta_I^i=\alpha\psi_I^i-a\left|\psi_I^i\right|^3\psi_I^i+\gamma a\left|\psi_I^i\right|^3\psi_I^i-b\left(\psi_I^i-\theta_\infty\right)\\
&\beta_E^i=\alpha\psi_E^i-a\left|\psi_E^i\right|^3\psi_E^i-b\left(\psi_E^i-\theta_\infty\right)
\end{aligned}
\tag{63}
$$

where the constants $\psi_I^i$ and $\psi_E^i$ represent the function $\psi^i$ at $\xi=1$ and at $\xi=1+\delta$, respectively.

The element $\psi^{i+1}$ may be represented as

$$
\psi^{i+1}=-\frac{\xi^2}{6}+\frac{1}{6}\left\{-\frac{6\psi_E^{i+1}(1+\delta)+(1+\delta)^3}{\delta}\left(\frac{1}{\xi}-1\right)+\frac{6\psi_I^{i+1}+1}{\delta}\left(\frac{1+\delta}{\xi}-1\right)\right\},\ 1\le\xi\le1+\delta,
$$
$$
i=0,1,2,3,\dots
\tag{64}
$$

It must be highlighted that, since $\psi^0\equiv0$, the constants $\psi_I^0$ and $\psi_E^0$ are zero too. The constants $\psi_I^{i+1}$ and $\psi_E^{i+1}$ are obtained from the boundary conditions of (62), as the solution of the following linear system (highlighting that $\beta_I^i$ and $\beta_E^i$ are known)

$$
\begin{aligned}
&-\frac{1}{3}+\frac{1}{6}\left\{\frac{6\psi_E^{i+1}(1+\delta)+(1+\delta)^3}{\delta}-\frac{6\psi_I^{i+1}+1}{\delta}(1+\delta)\right\}=\alpha\psi_I^{i+1}-\beta_I^i\\
&\frac{(1+\delta)}{3}-\frac{1}{6}\left\{\frac{6\psi_E^{i+1}+(1+\delta)^2}{\delta}\left(\frac{1}{1+\delta}\right)-\frac{6\psi_I^{i+1}+1}{\delta}\left(\frac{1}{1+\delta}\right)\right\}=\alpha\psi_E^{i+1}-\beta_E^i\\
&\beta_I^i=\alpha\psi_I^i-a\left|\psi_I^i\right|^3\psi_I^i+\gamma a\left|\psi_I^i\right|^3\psi_I^i-b\left(\psi_I^i-\theta_\infty\right)\\
&\beta_E^i=\alpha\psi_E^i-a\left|\psi_E^i\right|^3\psi_E^i-b\left(\psi_E^i-\theta_\infty\right)
\end{aligned}
\tag{65}
$$

The constants $\psi_I^{i+1}$ and $\psi_E^{i+1}$ that satisfy the above system may be represented as

$$
\begin{aligned}
\psi_E^{i+1}&=\frac{\left\{6\beta_E^i(1+\delta)+2(1+\delta)^2\right\}(1+\delta+\alpha\delta)+6\beta_I^i-2-\alpha\left(\delta^2+2\delta\right)}{6\left(2\alpha+2\alpha\delta+\alpha^2\delta+\alpha\delta^2+\alpha^2\delta^2\right)}\\
\psi_I^{i+1}&=\frac{\left(3\delta+3\delta^2+\delta^3\right)+3\beta_E^i(1+\delta)^2+3\beta_I^i}{3\alpha}-(1+\delta)^2\psi_E^{i+1}
\end{aligned}
\tag{66}
$$

In order to estimate a sufficiently large (not necessary) value for $\alpha$, we could use (52) as follows

$$
v=-\frac{1}{6}\xi^2\ \Rightarrow\ \sup_\Omega v=0,\ \inf_\Omega v=-\frac{1}{6}(1+\delta)^2\ \Rightarrow
$$
$$
\Rightarrow\ \sup_\Omega(\theta)\le\left\{\frac{b\theta_\infty}{a(1-\gamma)}+\frac{(1+\delta)}{3a(1-\gamma)}\right\}^{1/4}+\frac{1}{6}(1+\delta)^2\ \Rightarrow
$$
$$
\Rightarrow\ \alpha\ge4a\left\{\left\{\frac{b\theta_\infty}{a(1-\gamma)}+\frac{(1+\delta)}{3a(1-\gamma)}\right\}^{1/4}+\frac{1}{6}(1+\delta)^2\right\}^3+b
\tag{67}
$$

but this will give rise to values much greater than the necessary.

Table 1 presents $\theta_I$ and $\theta_E$ for some selected values of $\gamma$, $\delta$, $a$, $b$, and $\theta_\infty$. The non-convexity effect is present in lines where $\gamma\neq0$. The results obtained with $\gamma=0$ disregard the effect of the non-convexity.

It is worthnotingthat the temperature increase due to non-convexity effects may reach 50%.

Figures 3 and 4 present the dimensionless temperature distribution for two values of $\delta$ ($\delta=1.0$ and $\delta=2.0$) and some selected values of $a$, $b$, and $\theta_\infty$. The results were obtained for $\gamma=0.8$ and $\gamma=0$ (without the non-convexity effects).

Figure 5 presents $\psi_I^i$ and $\psi_E^i$ (elements of the sequence) as a function of $i$ for three considered values of $\alpha$, illustrating the process of convergence.

From Figures 3 and 4, the effects of non-convexity on the temperature distributions is evident:the non-convexity gives rise to a non-negligible temperature increase.

It is interesting to notice that, if the convergence is reached for $\alpha = 10.0$, the convergence is ensured for any $\alpha \geq 10.0$. Nevertheless, as $\alpha$ increases, the speed of convergence decreases. If $\alpha = 10.0$, the element $\psi_I^{10}$ is considered a good approximation (error less than 1%). On the other hand, if $\alpha = 30.0$, the element $\psi_I^{10}$ is not a good approximation (more elements are required).

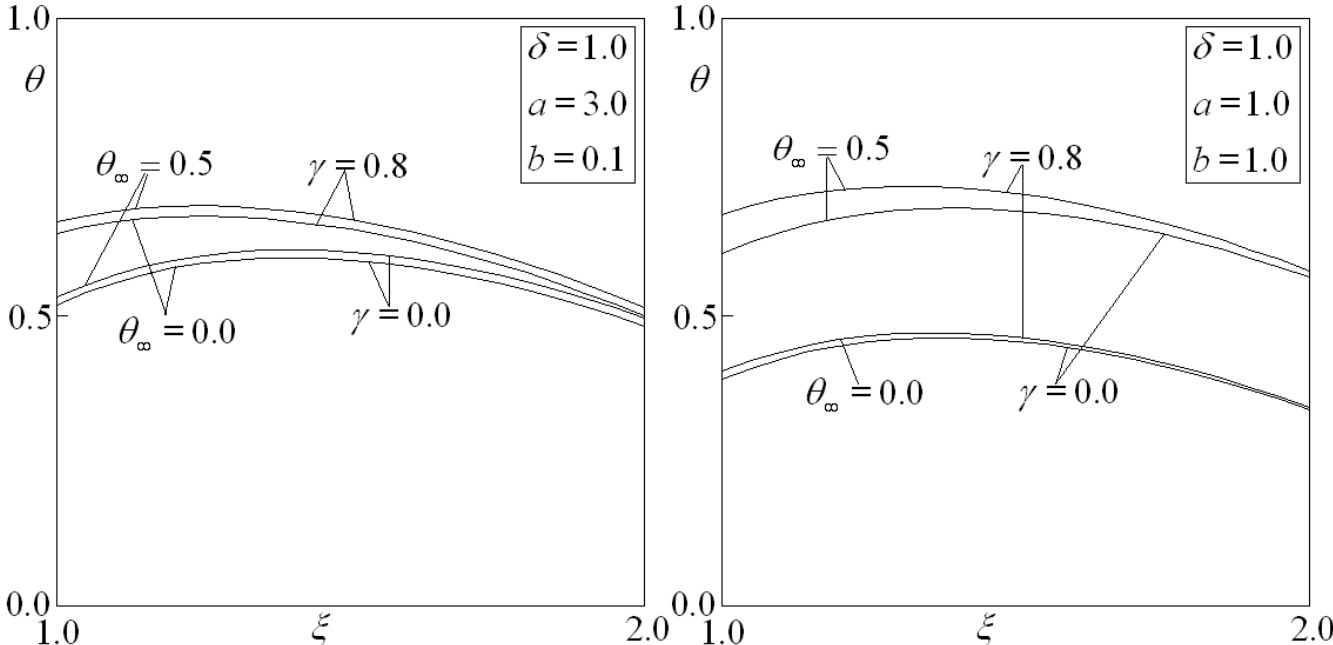

**Figure 3.** The dimensionless temperature $\theta$ as a function of the dimensionless position $\xi$ for $\delta = 1.0$.

**Table 1.** Some values of $\theta_I$ and $\theta_E$ obtained for 16 selected cases.

| | | | | | | |
|---|---|---|---|---|---|---|
| $\gamma = 0.0$ | $\delta = 0.1$ | $a = 1.0$ | $b = 0.1$ | $\theta_\infty = 0.8$ | $\theta_I = 0.52449$ | $\theta_E = 0.52434$ |
| $\gamma = 0.8$ | $\delta = 0.1$ | $a = 1.0$ | $b = 0.1$ | $\theta_\infty = 0.8$ | $\theta_I = 0.57611$ | $\theta_E = 0.57176$ |
| $\gamma = 0.0$ | $\delta = 0.5$ | $a = 1.0$ | $b = 0.1$ | $\theta_\infty = 0.8$ | $\theta_I = 0.71713$ | $\theta_E = 0.70530$ |
| $\gamma = 0.8$ | $\delta = 0.5$ | $a = 1.0$ | $b = 0.1$ | $\theta_\infty = 0.8$ | $\theta_I = 0.81699$ | $\theta_E = 0.75004$ |
| $\gamma = 0.0$ | $\delta = 0.5$ | $a = 2.0$ | $b = 1.0$ | $\theta_\infty = 0.8$ | $\theta_I = 0.66748$ | $\theta_E = 0.65842$ |
| $\gamma = 0.8$ | $\delta = 0.5$ | $a = 2.0$ | $b = 1.0$ | $\theta_\infty = 0.8$ | $\theta_I = 0.75259$ | $\theta_E = 0.68234$ |
| $\gamma = 0.0$ | $\delta = 1.0$ | $a = 1.0$ | $b = 0.1$ | $\theta_\infty = 0.4$ | $\theta_I = 0.84847$ | $\theta_E = 0.79670$ |
| $\gamma = 0.8$ | $\delta = 1.0$ | $a = 1.0$ | $b = 0.1$ | $\theta_\infty = 0.4$ | $\theta_I = 1.02156$ | $\theta_E = 0.82822$ |
| $\gamma = 0.0$ | $\delta = 1.0$ | $a = 1.0$ | $b = 0.1$ | $\theta_\infty = 0.8$ | $\theta_I = 0.86490$ | $\theta_E = 0.81461$ |
| $\gamma = 0.8$ | $\delta = 1.0$ | $a = 1.0$ | $b = 0.1$ | $\theta_\infty = 0.8$ | $\theta_I = 1.04694$ | $\theta_E = 0.84610$ |
| $\gamma = 0.0$ | $\delta = 0.5$ | $a = 2.0$ | $b = 1.0$ | $\theta_\infty = 0.8$ | $\theta_I = 0.66748$ | $\theta_E = 0.65842$ |
| $\gamma = 0.8$ | $\delta = 0.5$ | $a = 2.0$ | $b = 1.0$ | $\theta_\infty = 0.8$ | $\theta_I = 0.75259$ | $\theta_E = 0.68234$ |
| $\gamma = 0.0$ | $\delta = 1.0$ | $a = 1.0$ | $b = 0.5$ | $\theta_\infty = 0.4$ | $\theta_I = 0.78019$ | $\theta_E = 0.72717$ |
| $\gamma = 0.8$ | $\delta = 1.0$ | $a = 1.0$ | $b = 0.5$ | $\theta_\infty = 0.4$ | $\theta_I = 0.89469$ | $\theta_E = 0.74911$ |
| $\gamma = 0.0$ | $\delta = 3.0$ | $a = 1.0$ | $b = 0.5$ | $\theta_\infty = 0.4$ | $\theta_I = 1.22369$ | $\theta_E = 0.96425$ |
| $\gamma = 0.8$ | $\delta = 3.0$ | $a = 1.0$ | $b = 0.5$ | $\theta_\infty = 0.4$ | $\theta_I = 1.64805$ | $\theta_E = 0.97264$ |

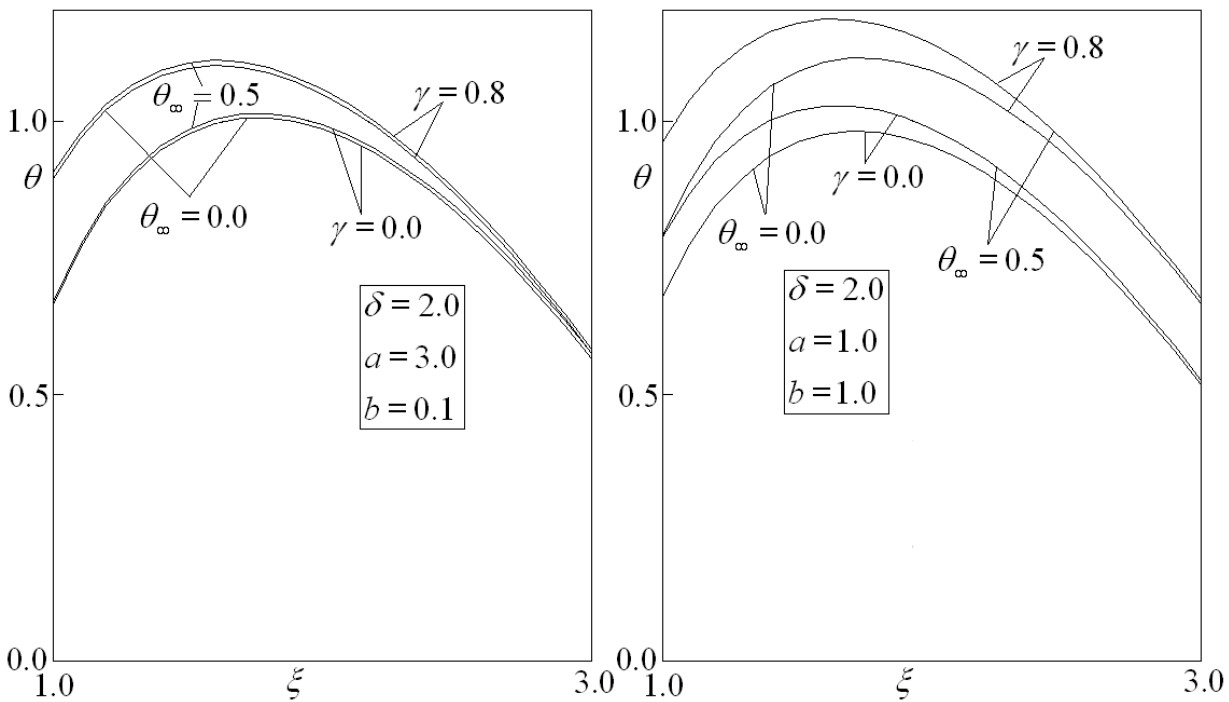

**Figure 4.** The dimensionless temperature $\theta$ as a function of the dimensionless position $\xi$ for $\delta = 2.0$.

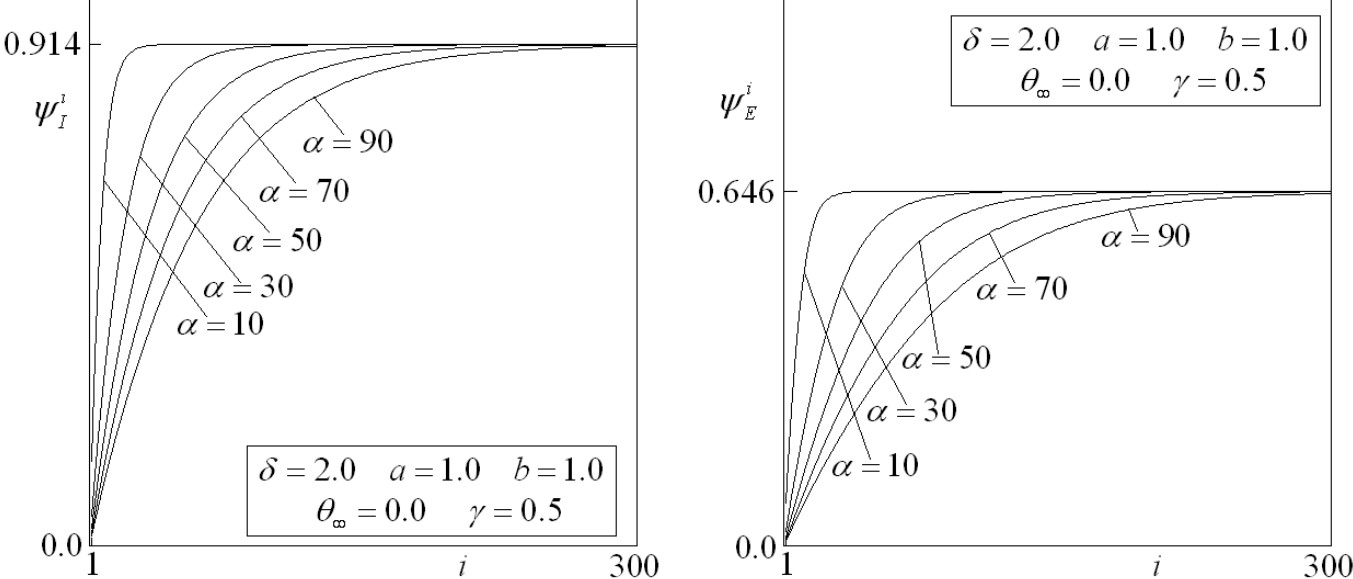

**Figure 5.** The constants $\psi_I^i$ and $\psi_E^i$ represented as a function of $i$ for five different values of the constant $\alpha$ for the particular case in which $\delta = 2.0$, $a = b = 1.0$, $\theta_\infty = 0.0$, and $\gamma = 0.5$. In this case, $\theta_I = 0.914$ and $\theta_E = 0.646$.

## 9. Conclusions

A very simple procedure for constructing the solution of a non-linear heat transfer problem was presented in this work. The proposed scheme usesmethods known to most undergraduate students.

The problem, which is inspired by conduction-convection-radiation heat transfer processes, consists of an interesting issue, usually considered under a severe simplifying hypothesis, in order to become mathematically simpler.

The main result of the paper is as follows. The nonlinear heat transfer problem

$$
\begin{aligned}
&div(k\nabla T) + \dot{q} = 0 \ in \ \Omega \\
&-k\nabla T \cdot \mathbf{n} = \sigma|T|^3 T - \int_{\partial\Omega} \left\{ \sigma|T|^3 T \right\} K dA - s + h(T - T_\infty) \ on \ \partial\Omega
\end{aligned}
\tag{68}
$$

is treated as a sequence of (well-known) problems such as

$$
\begin{aligned}
&div(k\nabla T) + \dot{q} = 0 \ in \ \Omega \\
&-k\nabla T \cdot \mathbf{n} = \alpha(T - T_\infty) \ on \ \partial\Omega
\end{aligned}
\tag{69}
$$

in which $\alpha$ is a known positive constant.

In this way, a process involving the effects of high temperatures is treatedas a basic undergraduate problem.

It is remarkable that the procedure used for constructing the solution can be extended toreaching approximate solutions. For instance, we could employ a discretized form of (11) in order to find an approximate limit for the sequence.

**Funding:** This work was supported by Brazilian Agency CNPq (Grant 304962/2022-8).

**Data Availability Statement:** Not applicable.

**Acknowledgments:** The author, R. M. S. Gama, gratefully acknowledges the support provided by Brazilian Agency CNPq (Grant 304962/2022-8) and by Brazilian Agency CAPES (Finance code 001).

**Conflicts of Interest:** The author declares no conflict of interest.

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
