# Peer review of "The Heat Transfer Problem in a Non-Convex Body—A New Procedure for Constructing Solutions"

_axioms, doi:10.3390/axioms12040338_

Round 1

Reviewer 1 Report

The heat transfer problem in a nonconvex body is solved in this paper. The technical contents are rich and the paper is well prepared. Some comments:

Due to the time limit of review, I am unable to check the correctness of each formula. Authors are suggested to have a careful proof. Besides, all variables should be clearly defined at their first appearance. For instance, I cannot find the definition of T in eq. (1).

I am curious whether there are early existing works on heat transfer in a nonconvex body. I cannot tell from the current literature review, as only a small amount of technical papers are discussed. Journal papers within the recent 5 years may be supplemented. Besides, the merits of this finding can be highlighted if you could mention its application in industrial systems such as heating system operation in 10.1016/j.energy.2018.01.049 and 10.1109/TSG.2022.3210014, but note that this is not mandatory. 

Following the previous comment, in engineering practice, we usually use a discretized solution rather than a continuous one. Can you make a tentative remark on how your solution may be applied to dispatch in discrete time intervals?

Seems like the solution depends on the parameter /alpha. How to tune /alpha?

Author Response

Dear Reviewer 1,

            Thank you very much for your comments and suggestions.

            In fact, (except articles from myself), there is no articles considering the coupled conduction-radiation in nonconvex bodies.

            Answering the last remark, the solution is reached for any value of the parameter “alpha”, provided (21) holds. In other words, if alpha=1.0 ensures convergence, then any alpha>1.0 will ensure too. When alpha=0.2 ensures convergence, then alpha=10000000 will ensure convergence too.

            About discretized solutions, I have inserted a “tentative remark” in the conclusions.

            Both the suggested references were added.

            I have inserted the definition of  T (equation (1).

Reviewer 2 Report

In the paper under review, the author investigates the coupled steady-state conduction-radiation-convection heat transfer phenomenon in a non convex black body, which is represented by a second order partial differential equation with nonlinear boundary conditions. A procedure is proposed for constructing a solution of the corresponding problem by means of a sequence whose elements are obtained from some linear problems. Of course, the topic of work, the results and the proposed approaches are very interesting. However, there are some questions and comments.

1) In what function space is the solution of the problem sought? Maybe this is the Sobolev space?

2) In what sense is the convergence (limit) in formula (12) understood?

3) Does any version of the maximum principle hold for the problem under consideration?

4) What does the term dA mean?

5) A more detailed review of the current literature on nonlinear heat transfer problems is needed.

Author Response

Dear Reviewer 2,

            In fact, we consider the Sobolev Space H1. Since the source is bounded, the solution will belong to H2. Taking into account the properties (physical and geometrical), the solution T is continuous in omega.

            In this way, the limit is valid inside omega.

            The maximum principle holds for this problem. The maximum of the difference in omega (as well as the minimum) takes place on the boundary (see (14)).

            The “dA” denotes an infinitesimal surface element. This is usually used in Mechanics.

            I have added 5 references and inserted a paragraph in Introduction. There is a lack of articles in this field.

Round 2

Reviewer 1 Report

Thanks for the revision. The authors have addressed all my concerns and the paper is acceptable to me.

Author Response

Again, I would like to acknowledge to Reviewer 1.

Reviewer 2 Report

Overall, I am satisfied with the answers of the author for my comments. However, I believe that the manuscript needs to be further improved before publication. On the site of the journal  "Axioms", it is mentioned that "Axioms is an international, peer-reviewed, open access journal of mathematics, mathematical logic and mathematical physics". Therefore, the author should devote more to the mathematical aspects of their research in order to attract the attention of the target audience of this journal. Additional comments in the paper should be made on this issue. In particular, it is necessary to explicitly describe the functional spaces used, the norms in them, as well as the type of convergence of the sequence of approximate solutions to a solution of the original problem.  In general, taking into account the choice of the journal "Axioms", it is best to format the obtained results in the form of theorems and lemmas. Moreover, to emphasize the importance of the research of heat transfer problems with nonlinear boundary conditions, I recommend making references to the recent works on this topic:
1) Domnich, A.A.; Baranovskii, E.S.; Artemov, M.A. A nonlinear model of the non-isothermal slip flow between two parallel plates. J. Phys. Conf. Ser. 2020, 1479, 012005.
2) Baranovskii E.S. Optimal boundary control of the Boussinesq approximation for polymeric fluids. J. Optim. Theory Appl. 2021, 189, 623-645.

Author Response

First of all, reviewer 2 states that he is “satisfied with the answers” regarding the first round of comments.

Although reviewer 2 generated new requests, some of them involves only subjective points of view. There is only one item that the reviewer 2 stats as “must be improved” (solved with the new suggested references).

In addition, the article was improved as follows:

  • The functional spaces and the norms were presented. Some of them were added in this new version of the paper.
  • The type of convergence is explicitly shown in the (new) equation (55)
  • The two recommended recent works (from Baranovskii) were cited.

The comments of reviewer 2 are the following (in bold face we have the main points).

Overall, I am satisfied with the answers of the author for my comments. However, I believe that the manuscript needs to be further improved before publication. On the site of the journal  "Axioms", it is mentioned that "Axioms is an international, peer-reviewed, open access journal of mathematics, mathematical logic and mathematical physics". Therefore, the author should devote more to the mathematical aspects of their research in order to attract the attention of the target audience of this journal. Additional comments in the paper should be made on this issue. In particular, it is necessary to explicitly describe the functional spaces used, the norms in them, as well as the type of convergence of the sequence of approximate solutions to a solution of the original problem.  In general, taking into account the choice of the journal "Axioms", it is best to format the obtained results in the form of theorems and lemmas. Moreover, to emphasize the importance of the research of heat transfer problems with nonlinear boundary conditions, I recommend making references to the recent works on this topic:
1) Domnich, A.A.; Baranovskii, E.S.; Artemov, M.A. A nonlinear model of the non-isothermal slip flow between two parallel plates. J. Phys. Conf. Ser. 2020, 1479, 012005.
2) Baranovskii E.S. Optimal boundary control of the Boussinesq approximation for polymeric fluids. J. Optim. Theory Appl. 2021, 189, 623-645.

Round 3

Reviewer 2 Report

The author has made the necessary additions in the article. In my opinion, the new version of the paper is well-written. Now it is easy to follow the ideas as well as derivations of the work. The obtained results give an important contribution in the subject area.
Conclusion: I recommend the article in the current form for publication in Axioms.